# The Role of Muscle Mass Gain Following Protein Supplementation Plus Exercise Therapy in Older Adults with Sarcopenia and Frailty Risks: A Systematic Review and Meta-Regression Analysis of Randomized Trials

**DOI:** 10.3390/nu11081713

**Published:** 2019-07-25

**Authors:** Chun-De Liao, Hung-Chou Chen, Shih-Wei Huang, Tsan-Hon Liou

**Affiliations:** 1School and Graduate Institute of Physical Therapy, College of Medicine, National Taiwan University, Taipei 10055, Taiwan; 2Department of Physical Medicine and Rehabilitation, Shuang Ho Hospital, Taipei Medical University, New Taipei City 23561, Taiwan; 3Center for Evidence-Based Health Care, Shuang Ho Hospital, Taipei Medical University, New Taipei City 23561, Taiwan; 4Graduate Institute of Sports Science, National Taiwan Sport University, Taoyuan 33301, Taiwan; 5Department of Physical Medicine and Rehabilitation, School of Medicine, College of Medicine, Taipei Medical University, Taipei 11031, Taiwan

**Keywords:** sarcopenia, protein supplement, exercise training, lean body mass, physical function

## Abstract

Aging and frailty are associated with a high risk of lean mass (LM) loss, which leads to physical disability and can be effectively alleviated by protein supplementation (PS) and muscle strengthening exercise (MSE). In this study, the associations between LM gain and PS + MSE efficacy (measured using physical outcomes) in elderly patients with a high risk of sarcopenia or frailty were identified. A comprehensive search of online databases was performed to identify randomized controlled trials (RCTs) reporting the efficacy of PS + MSE in elderly patients with sarcopenia or frailty. The included RCTs were analyzed using meta-analysis and risk of bias assessment. We finally included 19 RCTs in this meta-analysis with a median (range/total) Physiotherapy Evidence Database score of 7/10 (5–9/10). The PS + MSE group exhibited significant improvements in the whole-body LM (standard mean difference (SMD) = 0.66; *p* < 0.00001), appendicular LM (SMD = 0.35; *p* < 0.00001), leg strength (SMD = 0.65; *p* < 0.00001), and walking capability (SMD = 0.33; *p* = 0.0006). Meta-regression analyses showed that changes in appendicular LM were significantly associated with the effect sizes of leg strength (β = 0.08; *p* = 0.003) and walking capability (β = 0.17; *p* = 0.04), respectively. Our findings suggest that LM gain after PS + MSE significantly contributes to the efficacy of the intervention in terms of muscle strength and physical mobility in elderly patients with a high risk of sarcopenia or frailty.

## 1. Introduction

Aging is associated with muscle attenuation, which may contribute to common characteristics of muscle weakness and impaired physical mobility observed in elderly individuals at high risks of sarcopenia and frailty [1,2,3]. In addition, the indices for classifying older adults as clinically having sarcopenia [4] or high frailty risk [5] have been established among which low muscle strength and poor physical performance, such as slow walking speed, are common risk factors. Therefore, the maintenance of muscle strength and the prevention of sarcopenia are extremely crucial to enable prefrail and frail elderly adults to successfully perform physical tasks because low levels of lean mass or appendicular skeletal mass are closely associated with physical difficulty and poor health status among elderly patients [6,7].

Various nutrient interventions, exercise therapies or a combination of both are advised to prevent sarcopenia or frailty in elderly individuals [8,9,10,11,12], among which protein supplement (PS) combined with muscle strengthening exercise (MSE) has been known to benefit lean mass gain and function enhancement in elderly individuals regardless of protein type and exercise protocol [11,13,14,15]. However, whether intervention-induced changes in muscle mass contribute to strength gain and physical mobility improvement after PS + MSE remains unclear. An individual with lean mass (LM) gain exhibits improved physical performance, and several previous meta-analysis studies have reported that an increase in LM is accompanied by significant strength gain [14,16,17,18,19] or improvements in physical functioning [14,18] after PS + MSE; however, other authors have reported conflicting results of such synergetic improvements in LM and strength [13,20] or physical function [17,19]. Given that low muscle mass is a well-established factor associated with strength loss and mobility limitations in elderly populations [7,21] and that sarcopenia is associated with suppressed muscle protein turnover and homeostasis [22,23], identifying the effects of muscle mass changes in response to PS + MSE on strength gains and physical improvements can help clinical practitioners to efficiently make clinical decisions and set appropriate intervention strategies for older populations with sarcopenia or frailty.

Previous systematic reviews and meta-analyses have investigated the effects of PS + MSE on either sarcopenic or frail elderly populations; however, the combined meta-analysis approach in elderly adults with sarcopenia and frailty has not yet been confirmed. This study examined the combined effects of PS + MSE in elderly adults who have high risks of sarcopenia and frailty. In addition, meta-regression was used to determine whether LM gain in response to PS + MSE exerted any effect on the intervention outcomes of strength and physical mobility.

## 2. Method

### 2.1. Design

The present study was conducted by following the guidelines recommended by the Preferred Reporting Items for Systematic Reviews and Meta-Analysis [24]. The protocol for this study was registered at PROSPERO (registration number: CRD42018109176). The study was carried out based on a comprehensive electronic search from online sources. The articles were obtained from online database, including PubMed, EMBASE, the Cochrane Library Database, the Physiotherapy Evidence Database (PEDro), China knowledge resource integrated database, and Google Scholar databases. Secondary sources included papers cited by articles retrieved from the abovementioned sources. No limitation was imposed on the publication year and language to minimize publication and language bias. Two authors (CDL and HCC) independently searched for relevant articles, screened them, and extracted data. Any disagreement between the authors were resolved through a consensus in which the other team members (THL and SWH) acted as arbitrators.

### 2.2. Search Strategy

Keywords used for participant conditions were: “older/elderly” OR “frailty/frail” OR “sarcopenia”. Keywords used for intervention were: “exercise training” AND “protein/amino-acid/nutrient supplement”. The detailed search formulas for each database were presented in online Appendix A.

### 2.3. Selection Criteria of Studies

Trials were included if they met the following criteria: (1) the study design was a randomized control trial (RCT); (2) experimental groups received PS (including adequate protein-based diet) plus MSE; (3) control groups received a placebo supplement, PS alone, MSE alone, or none of above; (4) exercise types included resistance training or a multicomponent exercise regime that consisted of MSE, aerobic exercise, balance training, and physical activity training; (5) the supplement intervention used protein sources including whey protein, leucine, casein, and soy, for consumption in isolation or combined with other nutrients (creatine, amino acids); (6) the study enrolled participants with mean age ≥ 60 years; the participants were hospitalized, institutionalized, or community-dwelling elderly individuals and with a high risk of sarcopenia or frailty and physical limitations. (7) the study reported the primary outcome measures of muscle mass or sarcopenia indices, including lean body mass (LBM), fat-free mass, appendicular lean mass (ALM), lean mass index, appendicular mass index, and skeletal mass index; and (8) the study reported the secondary outcomes, such as leg strength or physical function, including mobility and walking capability. Walking capability was measured using walk speed or walk endurance and was defined as 10-m walk time or 6-min walk distance.

Studies were eliminated if (1) the trial was conducted in vitro or in vivo in an animal model or if (2) the trial had a non-RCT design such as a case report, case series, or a prospectively designed trial without a comparison group.

### 2.4. Data Extraction

Data was extracted from each included trial and presented in an evidence table (Table 1) regarding: (1) characteristics of study design and sample (group design, gender, age); (2) characteristics of exercise training and PS; (3) measured time points; and (4) main outcome results. One author (C.-D.L.) has extracted the relevant data from included trials and the second author (S.-W.H.) checked the extracted data. Any disagreement between two authors was resolved by a consensus procedure. A third author (T.-H.L.) was further consulted if the disagreement persisted.

The trial parallels with PS plus MSE group were extracted as experimental groups and those with placebo supplement, PS alone, or MSE alone was extracted as control groups. If the trial had more than one experimental group or control intervention, each of the comparisons was served as an independent one for meta-analyses [25].

### 2.5. Assessment of Bias Risks and Methodological Quality of Included Studies

Quality assessment was performed using the PEDro quality score to assess the risk of bias. Methodological quality of all the included studies was independently assessed by two researchers in accordance with the PEDro classification scale, which is a valid measure of the methodological quality of clinical trials [26]. The PEDro scale scores 10 items including random allocation, concealed allocation, similarity at baseline, subject blinding, therapist blinding, assessor blinding, >85% follow up for at least one key outcome, intention-to-treat analysis, between-group statistical comparison for at least one key outcome, and point and variability measures for at least one key outcome. Each item is scored as either 1 for present or 0 for absent, and a total sum score ranging from 0 to 10 is obtained by summation of all the 10 items. On the basis of the PEDro score, the methodological quality of the included RCTs was rated as high (≥7/10), medium (4–6/10), and low (≤3/10) [27].

### 2.6. Data Synthesis and Analysis

We computed effect sizes for each study separately for primary and secondary outcome measures. The primary outcome measure as well as the secondary one was defined as a pooled estimate of the mean difference in change between the mean of the treatment (PS and resistance training) and the placebo (other-type supplement and resistance training) groups. If the exact variance of paired difference was not derivable, it was imputed by assuming a within-participant correlation coefficient of 0.98, 0.92, and 0.80 for lean body mass [28], muscle strength [29,30], and mobility [30,31], respectively, between the baseline and posttest measured data. If data were reported as median (range), they were re-calculated algebraically from the trial data to impute the sample mean and SD [25,32]. All the extracted outcome data were calculated as standard mean difference (SMD) versus placebo or active control, as well as the secondary outcomes including functional mobility. We used SMD for meta-analysis when different scales were used to measure the same concept (e.g., pain, function score).

Fixed effect or random effect models were used, depending on the existence of heterogeneity. Statistical heterogeneity was assessed using the *I*^2^ statistic and was estimated for significance (*p* < 0.05) and χ^2^ and F values greater than 50% [33]. A fixed effect model was used unless statistical heterogeneity was significant (*p* < 0.05), after which a random effects model was used.

The duration of follow up (FU) was assessed and defined as immediate (<3 months), short term (≥3 months, <6 months), medium term (≥6 months, <12 months), and long term (≥12 months).

Subgroup analysis was conducted by using methodological quality level, duration of intervention, participant types (i.e., community-dwelling patient or institutionalized resident), and conditions (i.e., sarcopenia, frailty, or others), exercise types (i.e., resistance training or multicomponent exercise regime), PS dose (i.e., <20 g/day or ≥20 g/day [34]), and types of control group (i.e., placebo, PS alone, or exercise training alone) in the included trials. All subgroup differences were tested for significance and an *I*^2^ statistics statistic was also computed in order to estimate the degree of subgroup variability. Potential publication bias was investigated using visual inspection of a funnel plot to explore possible reporting bias [35] and was assessed by the Egger’s regression asymmetry test [36] using the SPSS, Version 20.0, statistical software (IBM, Armonk, NY, USA). A value of *P* less than 0.05 was considered to be statistically significant. All analyses were conducted using RevMan 5.3 (The Nordic Cochrane Centre, Copenhagen, Denmark).

To assess the association between muscle mass gain and clinical outcomes (strength and mobility), an inverse-variance weighted meta-regression model was established with percent muscle mass gain as the independent variable and SMD for strength and mobility as dependent variables; the analysis was controlled for age, methodological design, and follow-up duration. If the trial had more than one experimental or control intervention, each comparison was performed independently for meta-regression analysis.

## 3. Results

### 3.1. Trial Flow

Figure 1 shows a flowchart of the selection processes. The final sample consisted of 19 RCTs [31,37,38,39,40,41,42,43,44,45,46,47,48,49,50,51,52,53,54], which were published between 1994 and 2019. A sample consisting of a total of 1888 participants with a mean (range) age of 78.7 (64.0–89.2) years was enrolled. From the total sample, 738 patients received a protein-type supplement in combination with MSE, 556 received exercise training with or without placebo supplement, 209 received PS alone, and 385 received placebo supplement alone or no intervention.

### 3.2. Study Characteristics

Table 1 shows a summary of demographic data and study characteristics of the included RCTs. Fifteen RCTs enrolled community-dwelling elderly individuals with frailty, sarcopenia, or mobility limitation [31,37,38,39,40,41,42,44,47,48,50,51,52,53,54,55], whereas the remaining four enrolled institutionalized residents [43,45,46,49]. Mostly all the included RCTs employed an intervention period of 3–6 months [31,38,40,41,42,44,46,47,48,49,50,51,52,53,54]; however, three RCTs had short intervention periods of <3 months [37,43,45], and one had a long period of approximately 9 months [39]. With respect to the follow-up duration, all the 19 included RCTs reported a short-term or medium duration of <9 months; only one RCT had a long-term follow-up period of >12 months [45].

### 3.3. Protein Supplementation Characteristics

Protocols for PS are summarized in Table 1. The protocol for protein supplementation varied widely across included trials. Regarding the amount of protein, the majority of the included RCTs employed daily PS with amounts of extra protein ranging from 3.0 to 40.8 g/day. Three RCTs used PS of <10.0 g/day [31,46,47], two used PS of >30.0 g/day [43,49], and three RCTs provided supplements immediately after exercise on training days with amounts of extra protein ranging from 7.4 to 20.0 g/session [40,42,44].

### 3.4. Protocol of Exercise Training

A summary of protocols for MSE is presented in Table 1. Regarding the mode of exercise, seven RCTs used resistance exercise training [37,41,43,48,51,52,54], 11 RCTs used multicomponent exercise regime, and one used aerobic training with weighted walking [53]. One RCTs used a long-term exercise duration of 36 weeks (108 sessions) [39], whereas a medium-period treatment duration of 12–24 weeks (24–116 sessions) was used by 15 RCTs [31,38,40,41,42,44,46,47,48,49,50,51,52,53,54], and the other three RCTs used a short-period intervention of <12 weeks (16–30 sessions) [37,43,45].

### 3.5. Risk of Bias in Included Studies

The individual PEDro scores are listed in Table 2. Among the 19 RCTs, the methodological quality of 12 was high [37,38,39,40,41,43,45,47,48,50,51,54] and that of the other seven was medium [31,42,44,46,49,52,53], with a median PEDro score of 7/10 (range 5/10 to 9/10). The interrater reliability associated with the cumulative PEDro score was acceptable with an intraclass correlation coefficient of 0.93 (95% CI: 0.82–0.97). Of the 19 included RCTs, all employed random allocation, similarity at the baseline, between-group comparisons, and point estimates and variability; in addition, four employed concealed allocation, nine incorporated subject blinding, four incorporated therapist blinding, 10 incorporated assessor blinding, 17 had adequate follow-up, and 12 employed intention-to-treat analysis.

### 3.6. Effectiveness on Muscle Mass

Changes in LBM or fat-free mass after PS + MSE were reported by 18 RCTs (29 comparisons) [31,37,38,39,40,41,42,43,45,46,47,48,49,50,51,52,53,54], and changes in ALM were reported by 10 RCTs (19 comparisons) [31,37,41,42,47,48,50,51,52,53] (Table 1). Results of meta-analyses showed significant short-term (SMD = 0.71, *p* < 0.00001) and medium-term (SMD = 0.56, *p* = 0.02) effects on LBM as well as on ALM in favor of PS + MSE (Figure 2 and Appendix A). The evidence showed an overall effect on LBM with a significant SMD of 0.66 (95%CI: 0.41–0.91, *p* < 0.00001; *I*^2^ = 79%) favoring PS + MSE; similar results was observed in ALM (SMD = 0.40, 95%CI: 0.15–0.66, *P* = 0.002; *I*^2^ = 59%) (Figure 2 and Appendix A).

The results of subgroup analyses for LBM (Table 3) showed significant subgroup differences between participant types (*I*^2^ = 84.4%, *p* = 0.01), among participant conditions (*I*^2^ = 88.4%, *P* = 0.0002), and intervention periods (*I*^2^ = 70.8%, *p* = 0.03). The institutionalized elderly participants appeared to have significant effects on LBM with a greater SMD of 1.34 (*p* < 0.0001) than their community-dwelling peers (SMD = 0.44, *p* < 0.00001). The frail elderly participants were more likely to exhibit greater effects on LBM (SMD = 0.90, *p* < 0.00001) than their peers with sarcopenia (SMD = 0.44, *p* < 0.00001); similar results were observed for ALM.

### 3.7. Effectiveness on Muscle Strength and Physical Mobility Outcome

Changes in the handgrip and leg strength were reported by 6 RCTs (13 comparisons) [31,37,41,50,51,52] and 11 RCTs (23 comparisons) [31,39,41,42,43,47,48,49,51,52,54], respectively. Results of the meta-analysis showed significant combined effects on handgrip and leg strength with SMDs of 0.44 (95% CI: 0.26–0.62; *p* < 0.00001; *I*^2^ = 43%) and 0.65 (95% CI: 0.39–0.90; *p* < 0.00001; *I*^2^ = 62%) during an overall follow-up duration, respectively (Figure 2 and Appendix A).

The treatment effect of PS + MSE on physical function was assessed using several mobility tests, including walking capability by 18 RCTs (23 comparisons) [31,37,38,39,41,43,44,47,48,51,52], chair-rise test by seven RCTs (13 comparisons) [37,38,39,41,48,51,52], timed up-and-go by two RCTs (three comparisons) [38,48], stair-climb test by three RCTs (five comparisons) [38,39,43], short physical performance battery by three RCTs (three comparisons) [41,44,51], and single leg stance by one RCT (six comparisons) [52]. Significant effects favoring PS + MSE were observed on walking capability (SMD = 0.33, 95% CI: 0.14–0.52; *p* = 0.0006; *I*^2^ = 39%; Figure 2 and Appendix A). No significant effects were identified in other mobility (Figure 2 and Appendix A).

The results of subgroup analyses showed that the control group types exhibited effects on leg strength (*I*^2^ = 92.5%, *p* < 0.00001) and chair-rise scores (*I*^2^ = 70.1%, *p* = 0.04; Table 4). In addition, subgroup analyses for leg strength showed significant subgroup differences between participant types (*I*^2^ = 85.2%, *p* = 0.009) as well as among participant conditions (*I*^2^ = 88.7%, *p* = 0.0001) and intervention periods (*I*^2^ = 90.5%, *P* < 0.0001; Table 4). The institutionalized elderly participants exhibited a greater change in leg strength in response to PS + MSE with a greater SMD on leg strength by 1.02 (*p* < 0.00001) than the community-dwelled peers (SMD = 0.56, *p* = 0.0001). No other factor was found to affect subgroup heterogeneity for leg strength, walking capability, and chair-rise test (all *p* > 0.05) (Table 4).

### 3.8. Associations of Muscle Mass Change with Muscle Strength and Physical Function

To evaluate the association between muscle mass (i.e., LBM and ALM) and effect sizes of physical outcomes (i.e., leg strength and walking capability), four multivariate meta-regression models that pooled all time frames were established using age, methodological quality, and follow-up duration as covariates. The results of the meta-regression analyses showed that changes in LBM (β = 0.16, 95% CI: 0.04–0.29; *p* = 0.01; Figure 3) and ALM (β = 0.08, 95% CI: 0.04–0.13; *p* = 0.003; Figure 4) were significantly associated with SMDs of leg strength; the results further indicated that elderly individuals who responded to PS + MSE by an increase in LBM or ALM of >2.5% may have achieved a positive effect size of leg strength. In addition, a greater change in ALM significantly predicted a greater effect size of walking capability (β = 0.17, 95% CI: 0.01–0.33; *p* = 0.04; Figure 5); however, no significant association was observed between LBM gain and SMD of walking capability.

### 3.9. Side Effects and Compliance

No clinically relevant adverse events, side effects, or serious complications were reported after exercise training or protein supplementation in the RCTs. The compliance of resistance-based and multicomponent-based MSE was reported as 84%–100% by six RCTs [41,43,48,51,52,54] and 44%–81% by six RCTs [38,39,40,42,44,47], respectively (Table 1). The compliance of PS was reported as 44%–100% by 13 RCTs [37,38,39,40,42,43,44,45,48,50,51,52,53] (Table 1).

### 3.10. Publication Bias

Visual inspection of a funnel plot of increase in LBM, leg strength, and walking capability did not identify substantial asymmetry (Figure 6). The Egger’s linear regression test for LBM also did not indicate any evidence of obvious reporting bias among the comparisons (*t* = 1.28, *p* = 0.21) as well as leg strength (*t* = −0.71, *p* = 0.48) and walking capability (*t* = −1.17, *p* = 0.26).

## 4. Discussion

This study demonstrated that PS + MSE exerted overall significant effects on muscle mass (LBM, ALM), muscle strength, and physical mobility in elderly people with high risks of sarcopenia and frailty, regardless of follow-up duration, participant type, exercise type, and type of control group. The results of this study also indicated that muscle mass gains (i.e., increases in LBM or ALM) are significantly associated with improvements in physical outcomes, particularly leg strength and walking capability.

In this meta-analysis, results of subgroup analyses based on control types showed that PS + MSE had greater effects on LBM, leg strength, and walking capability than did MSE-alone control. These results are consistent with the findings of our previous studies, which have indicated that additional PS augments LBM gain and strength gain during resistance training in elderly adults [19,56]. Consistent with previous reviews [57,58] and following the recommendations from the European Society for Clinical Nutrition and Metabolism Expert Group [59], the results of current meta-analysis supported the urgent need for elderly patients with a risk of sarcopenia or frailty to incorporate protein-based nutrition intervention and MSE to prevent the functional decline, particularly institutionalized residents who are at high risk of insufficient protein intake and physical inactivity [60,61,62,63].

PS in combination with resistance-type MSE has been identified as an efficient intervention for LM and strength gain in elderly individuals [11,13,15,19,59,64]. However, an intensity as high as 80%–95% one repetition maximum has been recommended for resistance-type MSE to induce maximal muscle hypertrophy or muscle fiber adaptation [65,66]; this intensity is not permissible for most frail elderly individuals, particularly those with cardiopulmonary dysfunction or physical limitations. Therefore, multicomponent exercise, which incorporates MSE with balance training, aerobic training, and functional activity (i.e., walking) are recommended for elderly patients to improve physical function and prevent fall [58,67,68]. In this study, the results of subgroup analysis based on exercise types showed that PS and multicomponent exercise had significant effects on LBM and ALM as well as PS and resistance exercise, which indicated that elder patients with sarcopenia or frailty responded favorably to a combination of PS and multicomponent exercise in reversing or preventing muscle mass loss.

Previous systemic reviews have shown nonsignificant effects on changes in muscle mass [20,69], muscle strength [20], and physical mobility [19] in response to PS + MSE for elderly adults who mostly were healthy or not frail. In this meta-analysis, we obtained conflicting results showing that PS + MSE is beneficial for LM and strength gain in an elderly population with high risks of sarcopenia and frailty; furthermore, we identified that institutionalized residents appeared to achieve greater effects on LBM and leg strength in response to PS + MSE than their community-dwelling peers. Different populations may explain the inconsistency between the results of previous reviews and the findings in the present meta-analysis, which further confirm the conclusion of previous authors indicating that individuals with sarcopenia or frailty may experience greater benefits in muscle mass gain and physical performance in response to PS + MSE than their healthy peers [15,70]. Therefore, targeting the sarcopenia or frailty indices in response to PS in combination with MSE may hold greater promise in the preservation of independence as well as the prevention of progress to frailty in the prefrail or frail elderly population.

Previous meta-analyses have observed that an increase in LM is accompanied by significant strength gain or function recovery after PS + MSE [14,16,17,18]. The results of meta-regression analyses in this study further confirmed previous results, which indicated that an increase in LM significantly predicts relatively greater strength gain or walking capability after PS + MSE. Furthermore, we identified that an increase of >2.0% to 3.0% in muscle mass predicts a positive effect of PS + MSE on leg strength and walking capability, which may explain the inconsistencies with other authors who reported conflicting results of such synergetic improvements in muscle mass and function [13,17,18,19,20].

Several limitations to our findings should be elucidated. First, based on the variation among protein supplement regimes (protein source, supplied amounts, timing of ingestion) and exercise regimes (training duration, training volume), endorsing a definite conclusion for the effect of specific type of PS or MSE on muscle mass or strength gains was difficult. Second, some of our included trials had small sample sizes [41,48]; the results of these studies that reflected no significant intervention effect on primary or secondary outcomes may have contributed negatively to the overall effect size. Finally, inadequate statistical power for subgroup analyses was noted. Several subgroups (such as intervention durations for ALM) included a small number of RCTs (less than six), which may not have adequate power for detecting differences among subgroups [71,72]; the results of such subgroup analyses should be cautiously interpreted.

## 5. Conclusions

This systematic review evidenced that PS incorporated with MSE is effective in promoting gain in muscle mass and strength and enhancing performance in physical mobility in elderly adults with a high risk of sarcopenia or frailty, compared with the placebo, PS-alone, or MSE-alone controls. In addition, muscle mass gains have effects on strength gain and function recovery, particularly the walking capability. Therefore, we concluded that PS in addition to resistance-type or multicomponent exercise may have extra effects to prevent or offset muscle loss and functional decline, particularly among elderly individuals who are frail community dwellers or institutionalized residents. The results of this study add knowledge about effective nutrients and exercise intervention strategies and an interdisciplinary practical approach to counteract muscle loss and functional decline in the elderly population. This is relevant for those working in geriatric care and rehabilitation settings such as clinical, hospitalized, institutionalized, and community settings. Based on limitations in our current study, additional studies with relatively large samples, as well as identification of specific supplementation protocols.

## Figures and Tables

**Figure 1 nutrients-11-01713-f001:**
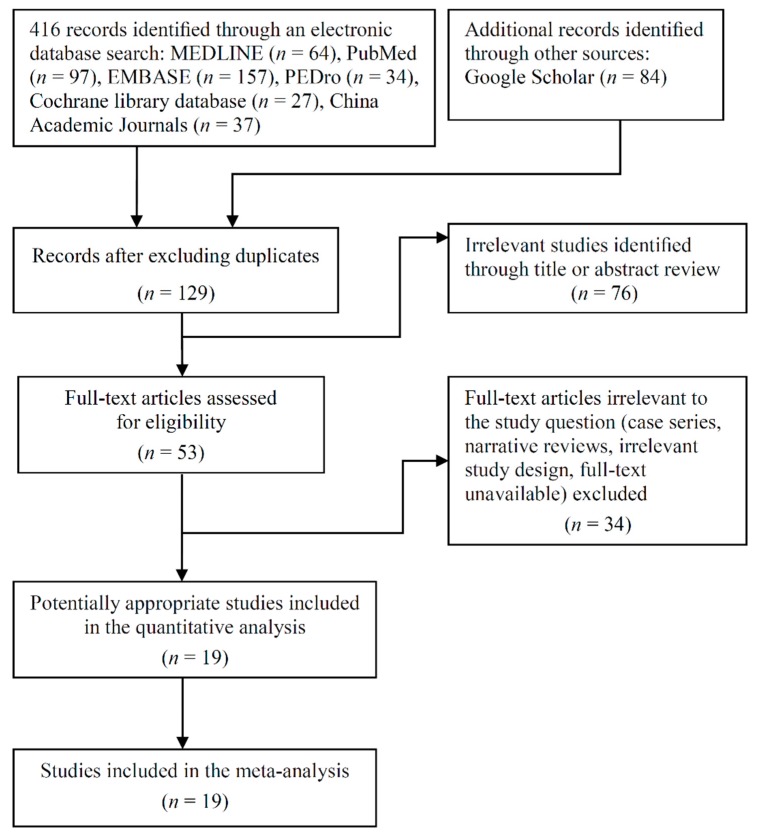
Flow chart of enrolled studies.

**Figure 2 nutrients-11-01713-f002:**
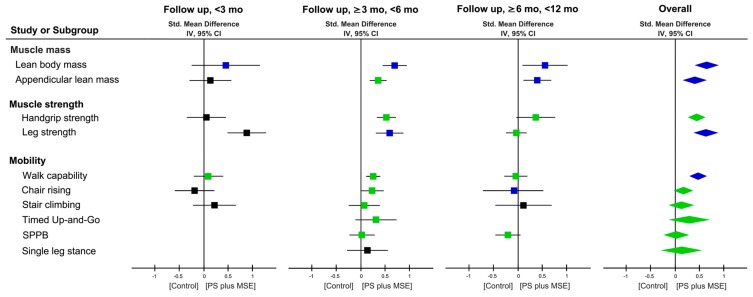
Forest plot summarizing effects of protein supplement (PS) plus muscle strengthening exercise (MSE) on changes of muscle mass, body composition, and physical function at each follow up duration. Each point estimate at each follow up duration (square) and during an overall duration (diamond) presents the combined effect (standard mean difference) of the outcome measure where indicated, with 95% CI (horizontal line). Results plotted on the right-hand side indicate effects in favor of PS plus Ex. The combined effects analyzed by a fixed- or random-effect model are denoted by green and blue colors, respectively; and a black colored square denotes that the combined effect is derived from a single study. 95% CI = 95% confidence interval; Std = standard; IV = inverse variance; SPPB = short physical performance battery.

**Figure 3 nutrients-11-01713-f003:**
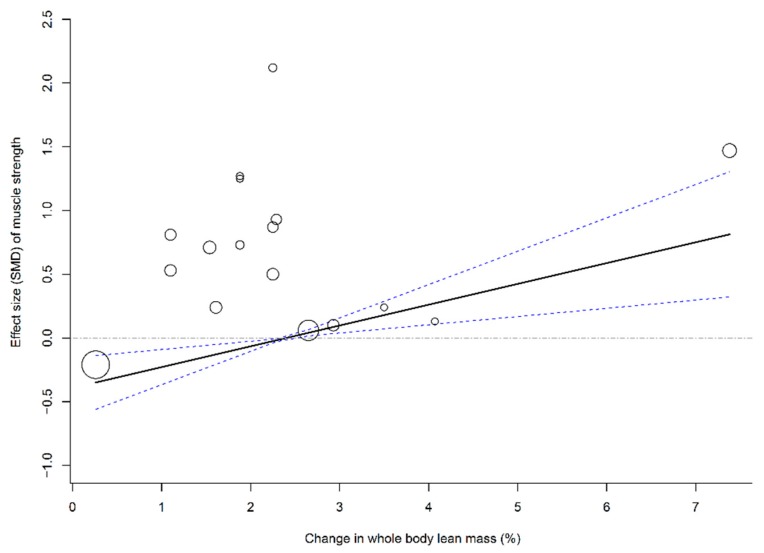
Multivariate meta-regression between percentage change in lean body mass and effects of PS plus MSE on leg strength. Each circle represents an independent comparison. The size of each circle is proportional to that study’s weight (inverse variance weighted). The regression prediction is represented by the solid line for effect size (SMD) of leg strength. Dotted lines represent the 95% CI. The metaregression model was adjusted for age, methodological quality, and follow-up time of each comparison. PS, protein supplementation; MSE, muscle strengthening exercise; SMD, standard mean difference.

**Figure 4 nutrients-11-01713-f004:**
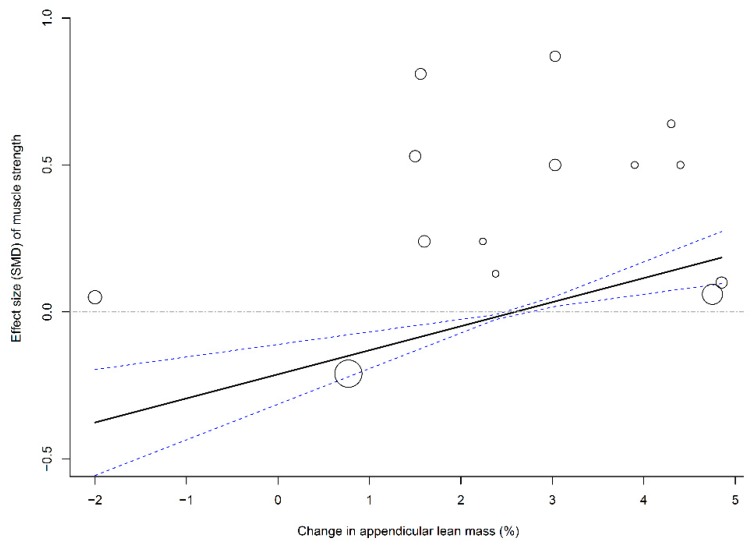
Multivariate meta-regression between percentage change in appendicular lean mass and effects of PS plus MSE on leg strength. Each circle represents an independent comparison. The size of each circle is proportional to that study’s weight (inverse variance weighted). The regression prediction is represented by the solid line for effect size (SMD) of leg strength. Dotted lines represent the 95% CI. The metaregression model was adjusted for age, methodological quality, and follow-up time of each comparison. PS, protein supplementation; MSE, muscle strengthening exercise; SMD, standard mean difference.

**Figure 5 nutrients-11-01713-f005:**
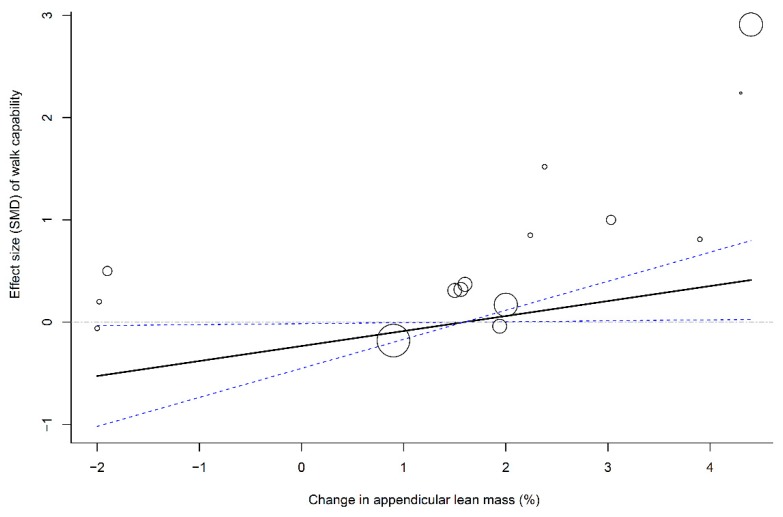
Multivariate meta-regression between percentage change in appendicular lean mass and effects of PS plus MSE on walk capability. Each circle represents an independent comparison. The size of each circle is proportional to that study’s weight (inverse variance weighted). The regression prediction is represented by the solid line for effect size (SMD) of walk capability. Dotted lines represent the 95% CI. The metaregression model was adjusted for age, methodological quality, and follow-up time of each comparison. PS, protein supplementation; MSE, muscle strengthening exercise; SMD, standard mean difference.

**Figure 6 nutrients-11-01713-f006:**
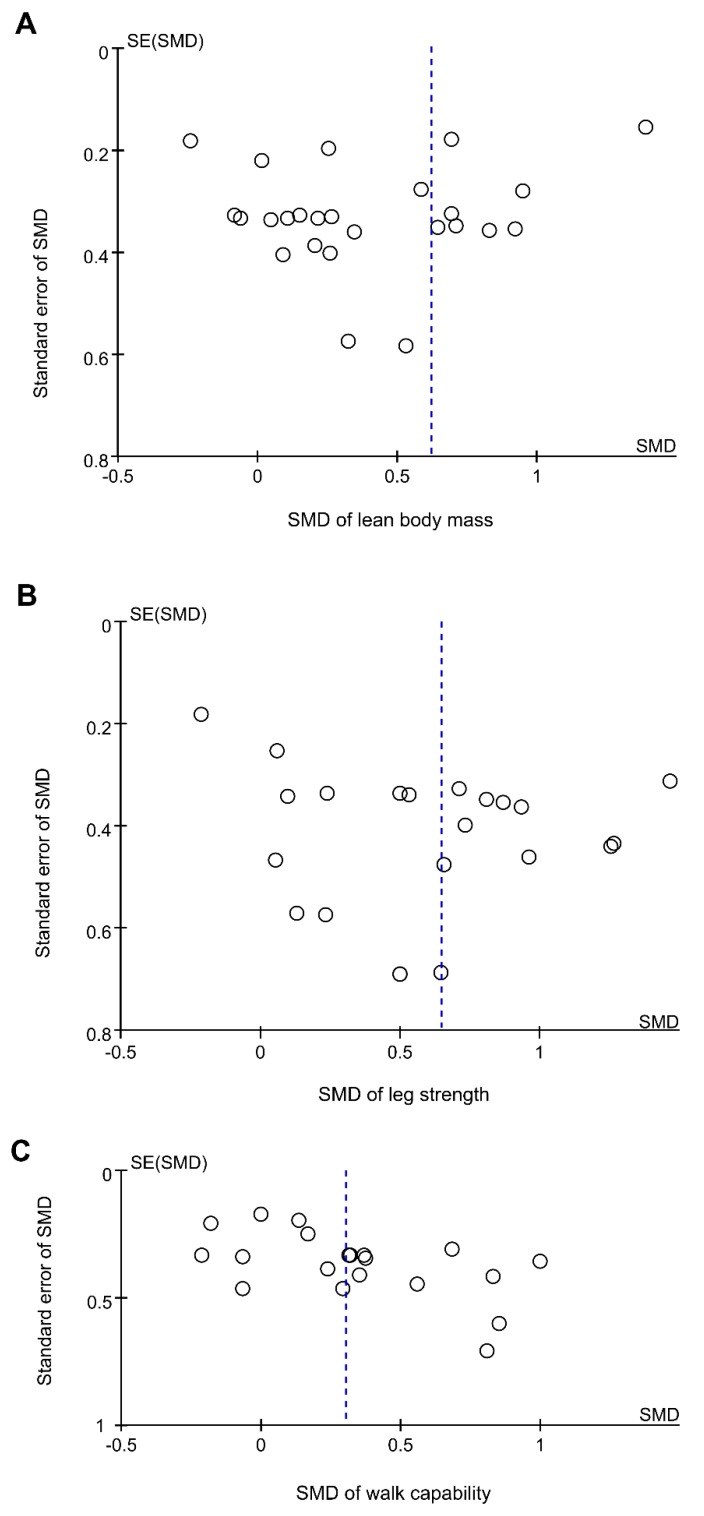
Funnel plots of the intervention effects for (**A**) lean body mass, (**B**) leg strength, and (**C**) walk capability. Each circle represents an independent comparison, with the *x*-axis representing standard mean difference (SMD) the over control comparisons and the *y*-axis showing the standard error (SE) of SMD. The vertical dotted line indicates the mean value of the SMDs.

**Table 1 nutrients-11-01713-t001:** Summary of included study characteristics.

Study (Author, Year, Ref)	Groups ^1^	Age (y) ^2^	Sex (F/M)	N	Design	Patient Type	Body Composition Assessment Method	Exercise Intervention		Protein supplement		Measured Time Point	Outcome Results
								Type, Compliance (%, EG/CG)	Frequency × Duration	Type, Compliance (%, EG/CG) ^10^	Intake Amount (g/d or g/session)		
Bjorkman, 2011 [37]	EG: MSE + PS	69.9 ± 7.4	20/3	23	RCT, DB	Sarcopenia	DXA	RET	2 d/wk × 8 wk	WP,	14.0 g/d	Baseline	↑ ALM/FM ratio ^6,7^
	CG: MSE	69.1 ± 6.9	22/2	24	Crossover	(PMR)		NA/NA	(16 sessions)	85.9/89.3		Posttest: 8 wk	↑ CRT ^6,7^; ↑ HG ^6,7^
Bonnefoy, 2012 [38]	EG: MSE + PS	86.3 ± 14.1 ^4^	49/4	53	RCT	Pre-frail elder	NA	MET	7 d/wk × 16 wk	Milk, soy, BCAA	20.0 g/d	Baseline	FFM ^8^; PASE ^8^; ADL ^8^; GS ^8^
	CG: Control *^3^*	86.0 ± 14.8 ^4^	39/10	49		adults		44/NA	(116 sessions)	44/NA		Posttest: 16 wk	↓ IADL ^5,7^; TUG ^8^
Bonnefoy, 2003 [39]	EG: MSE + PS	83.5 ± 1.2 ^9^	50/7 ^9^	57 ^9^	RCT, SB	Frail elder	DLW	MET	3 d/wk × 36 wk	Proteins	30.0 g/d	Baseline	FFM ^8^; GS ^8^; SC ^8^
	CG 1: MSE + PLA-S					individuals	method	63–70 ^9^	(108 sessions)	61/54		Midtest: 12 wk	↑ Leg strength ^7^
	CG 2: CT + PS											Posttest: 36 wk	
	CG 3: CT + PLA-S												
Carlsson, 2011 [40]	EG: MSE + PS	84.4 ± 6.3	33/9	42	RCT, DB	Frail	BIA	MET	2–3 d/wk × 13 wk	Milk protein	7.4 g/session	Baseline	ICW ^8^; BBS ^8^
	CG 1: MSE + PLA-S	85.3 ± 5.5	28/13	41		elderly		79/72	(29 sessions)	84/79		Midtest: 12 wk	
	CG 2: PS	82.7 ± 6.4	34/13	47								Posttest: 24 wk	
	CG 3: PLA-S	85.4 ± 7.2	36/11	47									
Dirks, 2017 [41]	EG: MSE + PS	76.0 ± 8.2	11/6	17	RCT, DB	Frail elderly	DXA	RET	2 d/wk × 24 wk	Milk protein	30.0 g/d	Baseline	↑ LBM ^6,^^7^; ↑ ALM ^6,^^7^
	CG: MSE + PLA-S	77.0 ± 8.2	11/6	17				84 ^9^	(48 sessions)	NR		Midtest: 12 wk	↓ CRT ^5,^^6^; ↑ SPPB ^5,^^6,8^
												Posttest: 24 wk	↑ LP 1-RM ^5,^^6^
Englund, 2017 [42]	EG: MSE + PS	78.1 ± 5.8	34/40	74	RCT, DB	Mobility-	DXA	MET	3 d/wk × 24 wk	Whey proteins	20.0 g/session	Baseline	↑ LBM ^5,6^; ↑ ALM ^5,6^
	CG: MSE + PLA-S	76.9 ± 4.9	35/40	75		limited elderly		>70	(72 sessions)	>85		Posttest: 24 wk	↑ Leg strength ^5,6^
Fiatarone, 1994 [43]	EG: MSE + PS	87.2 ± 6.0	16/9	25	RCT, DB	Nursing-	WBP	RET	3 d/wk × 10 wk	Soy protein	40.8 g/d	Baseline	WBP ^8^; ↑ GS ^6,^^7^; ↑ SC ^6,^^7^
	CG 1: MSE + PLA-S	86.2 ± 5.0	16/9	25		home residents	method	97/100	(30 sessions)	99/100		Posttest: 10 wk	↑ LP 1-RM ^6,^^7^
	CG 2: PS	85.7 ± 5.8	17/7	24									
	CG 3: PLA-S	89.2 ± 4.1	14/12	26									
Fielding, 2017 [44]	EG: MSE + PS	78.1 ± 5.8	34/40	74	RCT, DB	Mobility-	DXA	MET	3 d/wk × 24 wk	Whey proteins	20.0 g/session	Baseline	ALM
	CG: MSE + PLA-S	76.9 ± 4.9	35/40	75		limited elderly		75/72	(72 sessions)	88/86		Posttest: 24 wk	↑ GS ^5,^^6,8^; ↑ SPPB ^5,^^6,8^
Hegerova, 2015 [45]	EG: MSE + PS	83.6 ± 3.8	NR	100	RCT	Hospitalized	BIA	MET	6 d/wk × 3 wk	Protein	20.0 g/d	Baseline	SMM ^8^; HG ^8^
	CG: Control ^3^	83.2 ± 3.8		100		elderly adults		NR	(18 sessions)	83/71.3		Posttest: 3, 6, 12 mo	↓ Barthel index ^5,6,7^
Imaoka, 2016 [46]	EG: MSE + PS	87.6 ± 6.5	18/5	23	RCT	Institutionalized	BIA	MET	2 d/wk × 12 wk	Proteins	4.1 g/d	Baseline	SMI ^8^; FIM ^8^
	CG 1: MSE	82.6 ± 9.1	16/6	22		frail elderly		NR	(24 sessions)	NR		Posttest: 12 w	↓ Incidence of falls ^7^
	CG 2: PS	84.6 ± 7.7	20/3	23								Follow up: 26 wk	
	CG 3: Control *^3^*	82.5 ± 10.9	15/8	23									
Kim, 2016 [31]	EG: MSE + PS	80.9 ± 2.9	36/0	36	RCT	Sarcopenic	BIA	MET	2 d/wk × 12 wk	Leucine, EAA	3.0 g/d	Baseline	↑ LLM ^7^; ↑ GS ^7^; ↑ HG ^7^
	CG 1: MSE	81.4 ± 4.3	35/0	35		elderly		NR	(24 sessions)	NR		Posttest: 12 wk	↑ Leg strength ^7^
	CG 2: PS	81.2 ± 4.9	34/0	34		women							
	CG 3: Control *^3^*	81.1 ± 5.1	34/0	34									
Kim, 2012 [47]	EG: MSE + PS	79.5 ± 2.9	38/0	38	RCT, DB	Sarcopenic	BIA	MET	2 d/w × 12 w	Leucine, EAA	6.0 g/d	Baseline	↑ LBM ^6^; ↑ ALM ^6^
	CG 1: MSE	79.0 ± 2.9	39/0	39		elderly		70.3/71.8–80.5	(24 sessions)	NR		Posttest: 12 wk	↑ LLM ^6,^^7^; ↑ GS ^6,^^7^
	CG 2: PS	79.2 ± 2.8	39/0	39		women							↑ Leg strength ^6,7^
	CG 3: Control *^3^*	78.7 ± 2.8	39/0	39									
Maltais, 2016 [48]	EG 1: MSE + Milk PS	68.0 ± 5.6	0/8	8	RCT, DB	Sarcopenic	DXA	RET	3 d/w × 16 w	Milk protein,	19–20.5 g/d	Baseline	↑ LBM ^5,^^6^^,7^; ↓ TUG ^6,7^
	EG 2: MSE + EAA PS	64.0 ± 4.8	0/8	8		elderly men		>90 ^9^	(48 sessions)	EAA	(3.5 g leucine)	Posttest: 16 wk	
	CG: MSE + PLA-S	64.0 ± 4.9	0/10	10						>90 ^9^			
Molnar, 2016 [49]	EG: MSE + PS	66.6 ± 1.6	10/7	17	RCT	Institutionalized	BIA	MET	2 d/w × 12 w	WP, Leucine	33.0 g/d	Baseline	↑ LBM ^7^
	CG: MSE	66.4 ± 1.8	12/5	17		elderly		NR	(24 sessions)	NR		Posttest: 12 wk	↑ Leg strength ^7^
Rondanelli, 2016 [50]	EG: MSE + PS	80.8 ± 6.3	40/29	69	RCT, DB	Sarcopenic	DXA	MET	5 d/wk × 12 wk	WP	22.0 g/d	Baseline	↑ LBM ^6,^^7^; ↑ ADL ^5,^^6,7^
	CG: MSE + PLA-S	80.2 ± 8.5	37/24	61		elderly		NR	(60 sessions)	100 ^9^		Posttest: 12 wk	↑ HG ^6,^^7^; ↑ SF-36 PF ^7^
Tieland, 2012 [51]	EG: MSE + PS	78.0 ± 9.0	20/11	31	RCT, DB	Frail	DXA	RET	2 d/wk × 24 wk	Milk protein	30.0 g/d	Baseline	↑ LBM ^6,^^7^; ↑ ALM ^6,^^7^
	CG: MSE + PLA-S	79.0 ± 6.0	21/10	31		elderly		≥98 ^9^	(48 sessions)	≥98 ^9^		Midtest: 12 wk	↑ LP 1-RM ^5,^^6^; ↓ CRT ^5,^^6^
Yamada, 2019 [52]	EG: MSE + PS	84.9 ± 5.6	20/8	28	RCT, SB	Sarcopenic	DXA	RET	2 d/wk × 12 wk	WP	10.0 g/d	Baseline	↑ ALM ^6,^^7^; ↑ GS ^6,^^7^
	CG 1: MSE	84.7 ± 5.1	18/10	28		elderly		88.1/81	(24 sessions)	97.6/98.8–100		Posttest: 12 wk	↑ Leg power ^5,^^6,7^
	CG 2: PS	83.2 ± 5.7	20/8	28									↓ CRT ^5,^^6^; ↑ HG ^6,^^7^; SLS ^8^
	CG 3: Control^3^	83.9 ± 5.7	15/13	28									
Yamada, 2015 [53] *^11^*	EG: MSE + PS	78.1 ± 5.7	19/12	31	RCT	Frail	BIA	Weighted	7 d/wk × 24 wk	Protein (BCAA)	10.0 g/d	Baseline	↑ SMI ^7^
	CG 1: MSE	75.7 ± 5.8	8/7	15		elderly		walking		80 (67–92)		Posttest: 24 wk	
	CG 2: Control ^3^	76.4 ± 6.2	15/10	25				NR					
Zdzieblik, 2015 [54]	EG: MSE + PS	72.3 ± 3.7	0/26	26	RCT, DB	Sarcopenic	DXA	RET	3 d/wk × 12 wk	EAA	15.0 g/d	Baseline	↑ FFM ^6,7,8^
	CG: MSE + PLA-S	72.1 ± 5.5	0/27	27		Elderly men		86.7/90	(36 sessions)	NR		Posttest: 12 wk	

Note: ^1^ All parallels PS + MSE and control groups are presented for each trial. ^2^ Values are presented as mean and SD (or range). ^3^ Non supplement, non-exercise training, standardized care. ^4^ Data was estimated. ^5^ Significant within-group difference for control compared with baseline. ^6^ Significant within-group difference for PS + MSE compared with baseline. ^7^ Significant between-group difference for PS + MSE compared with control. ^8^ Non-significant between-group difference for PS + MSE compared with control. ^9^ Values of all samples. ^10^ Values denote the compliance of protein and placebo supplement (%) in EG and CG, respectively. ^11^ Only frail participants’ data were extracted. 6MWD, 6-min walk-for-distance; ADL, activities of daily living; ALM, appendicular lean mass; BBS, Berg’s balance scale; BCAA, branched chain amino acids; BIA, bioelectrical impedance analysis; BMI, body mass index; CG, control group; CRT, chair rise time; CSA, cross-sectional area; CT, cognition training; DB, double blind; DLW, doubly labeled water; DXA, dual-energy X-ray absorptiometry; EAA, essential amino acids; EG, experimental group; FIM, functional independence measure; FFM, fat-free mass; FFMI, fat-free mass index; GS, gait speed; HG, handgrip strength; ICW, intra cellular water; LBM, lean body mass; LLM, leg lean mass; LP 1-RM, leg press one repetition maximum; MET, multicomponent exercise training; MSE, muscle strengthening exercise; NA, not applicable; NR, not reported; PASE, Physical Activity Scale for older people; PLA-S, placebo supplement; PMR, polymyalgia rheumatica; PS, protein supplementation; RCT, randomized controlled trial; Ref = reference number; RET, resistance exercise training; RT, reaction time; SB, single blind; SC, stair climbing; SE, standard exercise; SF-36 PF, Short-Form 36-Item Health Survey (physical function subscore); SMI, skeletal muscle mass index; SMM, skeletal muscle mass; SLS, single leg stance; SPPB, short physical performance battery; TUG, timed up-and-go test; WBP, whole body potassium; WP, whey protein; ↑, significant increase; ↓, significant decrease. d: day; wk: week.

**Table 2 nutrients-11-01713-t002:** Summary of methodological quality based on the PEDro classification scale ^a^.

Study Author (year) (Reference Number)	Overall ^b^	Eligibility Criteria ^c^	1	2	3	4	5	6	7	8	9	10
Bjorkman 2011 [37]	8/10	*X*	*X*		*X*	*X*	*X*	*X*		*X*	*X*	*X*
Bonnefoy 2012 [38]	7/10	*X*	*X*	*X*	*X*				*X*	*X*	*X*	*X*
Bonnefoy 2003 [39]	7/10	*X*	*X*		*X*	*X*			*X*	*X*	*X*	*X*
Carlsson 2011 [40]	9/10	*X*	*X*	*X*	*X*	*X*	*X*	*X*	*X*		*X*	*X*
Dirks 2017 [41]	7/10	*X*	*X*		*X*	*X*			*X*	*X*	*X*	*X*
Englund 2017 [42]	6/10	*X*	*X*		*X*			*X*	*X*		*X*	*X*
Fiatarone 1994 [43]	8/10	*X*	*X*		*X*	*X*		*X*	*X*	*X*	*X*	*X*
Fielding 2017 [44]	6/10	*X*	*X*		*X*			*X*	*X*		*X*	*X*
Hegerova 2015 [45]	7/10	*X*	*X*	*X*	*X*				*X*	*X*	*X*	*X*
Imaoka 2016 [46]	6/10	*X*	*X*	*X*	*X*				*X*		*X*	*X*
Kim 2016 [31]	5/10	*X*	*X*		*X*				*X*		*X*	*X*
Kim 2012 [47]	7/10	*X*	*X*	*X*	*X*			*X*	*X*		*X*	*X*
Maltais 2016 [48]	8/10	*X*	*X*		*X*	*X*		*X*	*X*	*X*	*X*	*X*
Molnar 2016 [49]	6/10	*X*	*X*		*X*				*X*	*X*	*X*	*X*
Rondanelli 2016 [50]	9/10	*X*	*X*		*X*	*X*	*X*	*X*	*X*	*X*	*X*	*X*
Tieland 2012 [51]	7/10	*X*	*X*		*X*	*X*	*X*		*X*		*X*	*X*
Yamada 2019 [52]	6/10	*X*	*X*		*X*			*X*		*X*	*X*	*X*
Yamada 2015 [53]	6/10	*X*	*X*		*X*				*X*	*X*	*X*	*X*
Zdzieblik 2015 [54]	8/10	*X*	*X*		*X*	*X*		*X*	*X*	*X*	*X*	*X*
Summary ^#^		19	19	4	19	9	4	10	17	12	19	19

^a^ PEDro, Physiotherapy Evidence Database. Guidline of PEDro scale is available from PEDro database (https://www.pedro.org.au/english/downloads/pedro-scale/). ^b^ Points of methodological quality are denoted as “*X*” for fulfilled criteria. ^c^ Not used to calculate the total score. Score was determined by a third assessor. ^#^ This was calculated as the number of studies satisfied. PEDro classification scale: 1 = random allocation, 2 = concealed allocation, 3 = similarity at the baseline, 4 = subject blinding, 5 = therapist blinding, 6 = assessor blinding, 7 = more than 85% follow-up for at least one key outcome, 8 = intention-to-treat analysis, 9 = between-group statistical comparison for at least one key outcome, 10 = point and variability measures for at least one key outcome. Methodological quality: high, ≥7 points; medium, 4–6 points; low, ≤3 points.

**Table 3 nutrients-11-01713-t003:** Summary of overall effects and subgroup analyses results for muscle mass.

Subgroups	Lean Body Mass					Appendicular Skeletal Muscle Mass				
	Comparison, *n*	SMD	(95%CI)	*p*-value	*I*^2^ (%)	Comparison, *n*	SMD	(95%CI)	*p*-value	*I*^2^ (%)
Overall	29	0.66	(0.41, 0.91) ^‡^	<0.00001	79	19	0.35	(0.20, 0.50) ^‡^	<0.00001	59
MQ level										
PEDro score ≥7/10	19	0.52	(0.29, 0.75) ^‡^	<0.00001	64	9	0.63	(0.31, 0.95) ^‡^	0.0001	56
PEDro score <7/10	10	1.09	(0.41, 1.76) ^‡^	<0.00001	83	10	0.06	(−0.17, 0.29) ^†^	n.s.	39
Subgroup difference				n.s.	58.8				0.03	79.2
Participant type										
Community-dwelled	21	0.44	(0.24, 0.64) ^‡^	<0.0001	57	19	0.35	(0.20, 0.50) ^‡^	<0.00001	59
Institutionalized resident	8	1.34	(0.67, 2.01) ^‡^	<0.0001	84	0				
Subgroup difference				0.01	84.4				NA	NA
Participant condition										
Sarcopenia	11	0.44	(0.27, 0.62) ^†^	<0.00001	17	13	0.49	(0.14, 0.85) ^‡^	0.007	66
Frailty	17	0.90	(0.52, 1.27) ^‡^	<0.00001	82	2	0.75	(0.32, 1.18) ^†^	0.0006	0
Other conditions	1	−0.20	(−0.56, 0.16)	n.s.	NA	4	0.03	(−0.26, 0.33) ^†^	n.s.	0
Subgroup difference				0.0002	88.4				0.02	76.0
Gender										
Men	3	0.57	(0.12, 1.03) ^†^	0.01	0	2 ^a^	−0.03	(−0.82, 0.76) ^†^	n.s.	0
Women	8	0.34	(0.12, 0.56) ^†^	0.002	22	7	0.41	(−0.09 0.91) ^‡^	n.s.	77
Mixed	18	0.82	(0.46, 1.19) ^‡^	<0.0001	85	10	0.45	(0.13, 0.78) ^‡^	0.006	49
Subgroup difference				n.s.	53.6				n.s.	0
Control group type										
PLA-S or non-exercise	10	0.89	(0.42, 1.36) ^‡^	0.0002	90	5	0.80	(0.04, 1.56) ^‡^	0.04	89
Exercise	14	0.53	(0.21, 0.86) ^‡^	0.001	77	10	0.25	(−0.02, 0.52) ^‡^	n.s.	50
PS	5	0.65	(0.03, 1.27) ^‡^	0.04	85	4	0.53	(−0.16, 1.22) ^‡^	n.s.	55
Subgroup difference				n.s.	0				n.s.	5.4
Exercise type										
RET	11	0.57	(0.25, 0.88) ^‡^	0.0005	54	12	0.37	(0.13, 0.62) ^†^	0.002	42
MET	18	0.73	(0.38, 1.07) ^‡^	<0.0001	85	7	0.40	(0.01, 0.79) ^‡^	0.04	74
Subgroup difference				n.s.	0				n.s.	0
PS dose (g/day) ^b^										
<20	10	1.06	(0.43, 1.69) ^‡^	0.0009	87	7	0.24	(−0.08, 0.55) ^†^	n.s.	50
≥20	13	0.57	(0.24, 0.90) ^‡^	0.0008	78	6	0.37	(0.16, 0.58) ^†^	0.0005	28
Subgroup difference				n.s.	46.1				n.s.	0
Intervention duration (week)										
<12	4	0.14	(−0.18, 0.45) ^†^	n.s.	0	1	0.14	(−0.30, 0.58)	n.s.	NA
12–23	21	0.69	(0.42, 0.96) ^‡^	<0.00001	70	17	0.38	(0.09, 0.68) ^‡^	0.01	58
≥24	9	0.44	(0.05, 0.83) ^‡^	0.03	73	3	0.49	(−0.02, 1.00) ^†^	n.s.	65
Subgroup difference				0.03	70.8				n.s.	0

^†^ Fixed-model effect. ^‡^ Random-model effect. ^a^ Comparisons were derived from single trial. ^b^ Trials with protein supplement were included and those with amino-acid supplement were excluded. SMD, standard mean difference; I2, heterogeneity; MQ, methodological quality; PEDro, Physiotherapy Evidence Database; n.s., nonsignificant (*p* > 0.05); PLA-S, placebo supplement; PS, protein supplementation; RET, resistance exercise training; MET, multicomponent exercise training; NA, not applicable.

**Table 4 nutrients-11-01713-t004:** Summary of overall effects and subgroup analyses results for physical function.

Subgroups	Leg Muscle Strength					Walk Capability					Chair Rise				
	Comparison, *n*	SMD	(95%CI)	*p*-value	*I*^2^ (%)	Comparison, *n*	SMD	(95%CI)	*p*-value	*I*^2^ (%)	Comparison, *n*	SMD	(95%CI)	*p*-value	*I*^2^ (%)
Overall	23	0.65	(0.39, 0.90) ^‡^	<0.00001	62	23	0.33	(0.14, 0.52) ^‡^	0.0006	39	13	0.17	(−0.02, 0.37) ^†^	n.s.	0
MQ level															
PEDro score ≥ 7/10	12	0.79	(0.42, 1.16) ^‡^	<0.0001	66	13	0.24	(0.07, 0.41) ^†^	0.006	41	7	0.10	(−0.12, 0.32) ^†^	n.s.	9
PEDro score < 7/10	11	0.31	(0.09, 0.52) ^†^	0.005	40	10	0.25	(0.03, 0.48) ^†^	0.02	43	6^a^	0.47	(0.04, 0.91) ^†^	0.03	0
Subgroup difference				n.s.	50.5				n.s.	0				n.s.	55.7
Participant type															
Community-dwelled	19	0.56	(0.28, 0.85) ^‡^	0.0001	64	20	0.32	(0.11, 0.53) ^‡^	0.002	44	13	0.17	(−0.02, 0.37) ^†^	n.s.	0
Institutionalized resident	4	1.02	(0.62, 1.42) ^†^	<0.00001	0	3^a^	0.46	(0.00, 0.92) ^†^	0.05	0	0				
Subgroup difference				0.009	85.2				n.s.	0				NA	NA
Participant condition															
Sarcopenia	15	0.73	(0.44, 1.03) ^‡^	<0.00001	43	15	0.46	(0.15, 0.77) ^‡^	0.003	51	9	0.12	(−0.16, 0.40) ^†^	n.s.	0
Frailty	7	0.58	(0.32, 0.84) ^†^	<0.0001	51	7	0.27	(0.05, 0.49) ^†^	0.02	0	4	0.23	(−0.05, 0.50) ^†^	n.s.	24
Other conditions	1	−0.21	(−0.57, 0.15) ^‡^	n.s.	89	1	0.00	(−0.34, 0.34)	n.s.	NA	0				
Subgroup difference				0.0001	88.7				n.s.	49.3				n.s.	0
Gender															
Men	3	0.71	(−0.25, 1.67) ^‡^	n.s.	68	2^a^	1.15	(0.27, 2.03) ^†^	0.01	0	2 ^a^	0.15	(−0.65, 0.95) ^†^	n.s.	0
Women	7	0.79	(0.39, 1.20) ^‡^	0.0001	59	8	0.24	(0.03, 0.46) ^†^	0.02	45	2	0.03	(−0.31, 0.37) ^†^	n.s.	74
Mixed	13	0.51	(0.18, 0.83) ^‡^	0.002	54	13	0.21	(0.03, 0.39) ^†^	0.02	36	9	0.25	(−0.00, 0.51) ^†^	n.s.	0
Subgroup difference				n.s.	0				n.s.	41.7				n.s.	0
Control group type															
PLA-S or non-exercise	6	1.12	(0.67, 1.57) ^‡^	<0.00001	67	5	0.50	(0.29, 0.72) ^†^	<0.00001	56	4	0.52	(0.21, 0.83) ^†^	0.001	0
Exercise	12	0.43	(0.14, 0.73) ^‡^	0.004	63	15	0.25	(0.00, 0.50) ^‡^	0.05	48	7	0.00	(−0.24, 0.24) ^†^	n.s.	0
PS	5	0.69	(0.43, 0.96) ^‡^	<0.00001	56	3	0.33	(0.04, 0.61) ^†^	0.03	0	2^a^	0.31	(−0.22, 0.84) ^†^	n.s.	0
Subgroup difference				<0.00001	92.5				n.s.	0				0.04	70.1
Exercise type															
RET	14	0.61	(0.38, 0.83) ^†^	<0.00001	41	14	0.44	(0.11, 0.76) ^‡^	0.008	48	11	0.09	(−0.14, 0.32) ^†^	n.s.	0
MET	9	0.68	(0.25, 1.12) ^‡^	0.002	77	9	0.23	(0.06, 0.41) ^†^	0.01	27	2	0.41	(0.02, 0.79) ^†^	0.04	0
Subgroup difference				n.s.	0				n.s.	0				n.s.	48.5
PS dose (g/day) ^b^															
<20	7	0.86	(0.51, 1.22) ^†^	<0.00001	21	7	0.59	(−0.05, 1.23) ^‡^	n.s.	66	7	0.12	(−0.18, 0.41) ^†^	n.s.	13
≥20	10	0.48	(0.12, 0.84) ^‡^	0.01	63	10	0.28	(0.05, 0.50) ^†^	0.02	24	6	0.22	(−0.05, 0.48) ^†^	n.s.	0
Subgroup difference				n.s.	26.6				n.s.	0				n.s.	0
Intervention duration (week)															
<12	3^a^	1.06	(0.58, 1.54) ^†^	<0.0001	0	4	0.10	(−0.20, 0.41) ^†^	n.s.	44	1	−0.20	(−0.61, 0.21)	n.s.	NA
12–23	19	0.60	(0.32, 0.89) ^‡^	<0.0001	59	19	0.26	(0.11, 0.42) ^†^	0.0006	36	11	0.23	(−0.01, 0.48) ^†^	n.s.	0
≥24	4	−0.04	(−0.28, 0.21) ^†^	n.s.	0	4	−0.05	(−0.29, 0.19) ^†^	n.s.	57	3	−0.09	(−0.71, 0.52) ^‡^	n.s.	69
Subgroup difference				<0.0001	90.5				n.s.	2.7				n.s.	44

^†^ Fixed-model effect. ^‡^ Random-model effect. ^a^ Comparisons were derived from single trial. ^b^ Trials with protein supplement were included and those with amino-acid supplement were excluded. SMD, standard mean difference; I2, heterogeneity; MQ, methodological quality; PEDro, Physiotherapy Evidence Database; n.s., nonsignificant (*p* > 0.05); PLA-S, placebo supplement; PS, protein supplementation; RET, resistance exercise training; MET, multicomponent exercise training; NA, not applicable.

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
