# Peer review of "The Role of Muscle Mass Gain Following Protein Supplementation Plus Exercise Therapy in Older Adults with Sarcopenia and Frailty Risks: A Systematic Review and Meta-Regression Analysis of Randomized Trials"

_nutrients, 2019, doi:10.3390/nu11081713_

Round 1
Reviewer 1 Report
Liao and colleagues present results from a meta-analysis of studies examining the effect of protein supplementation in combination with exercise leads to greater increases/improvements in lean mass, leg strength, walk capability and chair rise compared to several different control groups in older adults.
The authors should be commended for the considerable amount of work required to collate and analyze the data contained in the current manuscript. While overall these results are of interest and generally well presented I do have the following comments/concerns:
1) In section 2.3 “selection criteria of studies” it is mentioned that only studies reporting results from subjects 60 yr or older are included but in Table 1 Arciero et al. 2014 studied subjects with a mean age of 47 yr, which is not an older population. Does 60 yr mean all subjects are older than this age, the mean is higher than 60 yr or just some subjects were older than this age? In my opinion it should be all were older than 60 yr. In this section number 4 is missing. Please also add a description of which types of exercise were included to this section.
2) The study would be cleaner and easier to interpret if only studies that examined effect of whole-proteins were included. For example, Kim et al 2012 (ref #52) used an amino acid supplement. If only protein studies were examined the effect of protein dose could be added to Tables 3 and 4.
3) Please exclude Kim et al 2015 (ref #51) as the supplement provided minimal protein (I calculate it to be 1.29 g/d) and was highly enriched in phospholipids.
4) Define walk capability/ability.
Author Response
[Nutrients]
Manuscript ID: nutrients-538306
Author's Reply to the Review Report (Reviewer 1)
Comments and Suggestions for Authors
Liao and colleagues present results from a meta-analysis of studies examining the effect of protein supplementation in combination with exercise leads to greater increases/improvements in lean mass, leg strength, walk capability and chair rise compared to several different control groups in older adults.
The authors should be commended for the considerable amount of work required to collate and analyze the data contained in the current manuscript. While overall these results are of interest and generally well presented I do have the following comments/concerns:
1) In section 2.3 “selection criteria of studies” it is mentioned that only studies reporting results from subjects 60 yr or older are included but in Table 1 Arciero et al. 2014 studied subjects with a mean age of 47 yr, which is not an older population. Does 60 yr mean all subjects are older than this age, the mean is higher than 60 yr or just some subjects were older than this age? In my opinion it should be all were older than 60 yr. In this section number 4 is missing. Please also add a description of which types of exercise were included to this section.
Response
Thank you for your comprehensive review and constructive comments. We removed the study Arciero et al. 2014 from the included studies and reperformed all analyses. The results are presented in the revised manuscript.
We have also added an item (number 4) in Section 2.3 to clarify the types of exercise that were included in this meta-analysis.
2) The study would be cleaner and easier to interpret if only studies that examined effect of whole-proteins were included. For example, Kim et al 2012 (ref #52) used an amino acid supplement. If only protein studies were examined the effect of protein dose could be added to Tables 3 and 4.
Response
According to the reviewer’s comment, we have added details on the subgroup analyses, which included only protein studies to examine the effect of protein dose, and the results are presented in Tables 3 and 4.
3) Please exclude Kim et al 2015 (ref #51) as the supplement provided minimal protein (I calculate it to be 1.29 g/d) and was highly enriched in phospholipids.
Response
We removed the study Kim et al 2015 from the included studies. Accordingly, all analyses were reperformed, and the results are presented in the revised manuscript.
4) Define walk capability/ability.
Response
We have defined walking capability in Section 2.3 as follows:
“(8) the study reported the secondary outcomes, such as leg strength or physical function, including mobility and walking capability. Walking capability was measured using walk speed or walk endurance and was defined as 10-meter walk time or 6-minute walk distance.”

Reviewer 2 Report
To Liao et al.,
The authors present a carefully performed meta-analysis and systematic review of the data examining the efficacy of protein supplementation and exercise in the improvement of muscle mass, muscle function, and physical function in older, sarcopenic adults. Whilst the study has been carefully analysed, there is a significant flaw within your study which has invalidated your findings.
In table 1, and throughout the results, you reference and use Arciero et al. (2014) as one of your 33 RCTs you have analysed. However, this study invalidates your selection criteria as not only are the participants of this study below the age of 60, they are classified as overweight or obese in the paper, with no reference to these adults as having sarcopenia and at a high risk of frailty. This study violates your inclusion criteria, and as such should not have been included. Therefore, the analyses, subgroup analyses and conclusions of your study are not valid and would need to be reperformed and reanalysed. However, as such a study was included without the participants being classified as sarcopenic or frail, I can no longer trust whether the remaining 32 articles are in keeping with the inclusion criteria.
Additionally, the rationale for the study is weak. You state that previous meta-analyses have examined protein supplementation and exercise in relation to sarcopenia, but do not make explicitly clear how your study is novel or different. In my mind, the novelty is the combined meta-analyses/systematic review approach and that you are examining (or should be examining) sarcopenic and frail older adults who had been enrolled in randomised controlled trials.
Finally, your text requires substantial editing for consistency of terminology (see below under minor), grammar, and usage of the English language.
Introduction
Paragraph 2
Line 2 – “Acute and chronic conditions” is vague. Do you mean with acute sarcopenia, chronic sarcopenia, or some other kind of condition?
Line 6 – Is “function” physical function or muscle function? I’d be clear here and throughout.
Line 5-8 – This section is unclear. You state that it is unclear whether a combination of PS and MSE has an effect on strength and function, yet in the following sentence, you list meta-analyses which have shown exactly that. Also, are you saying that these meta-analyses have examined muscle strength and physical functioning independently as per the “or” in line 8? If so, this needs to be made clearer.
This leads to my next point of novelty. If there are meta-analyses existing to have shown improvements in muscular strength and physical function in sarcopenic adults following PS and MSE, then how is your study novel? Reference 15 in particular appears to be very similar. Make it explicitly clear in your introduction how your study is different. The fact you have used RCT’s in adults with high sarcopenia and frailty risk makes this novel, but has not been made clear in the introduction.
General
As you are examining studies which have used older adults with a high sarcopenia or frailty risk, some indices of classifying older adults as clinically having sarcopenia or high frailty risk needs to be included in the introduction. The revised European Working Group on Sarcopenia consensus (EWGSOP2) would be a good place to start (Cruz-Jentoft et al., 2019; Age Ageing, 48[1], 16-31).
Method
Section 2.3
As you are examining MSE, the type of muscle strengthening exercise should eb considered in more detail in your inclusion criteria. In your results (section 3.4), you state exercise training, which is very broad and could encompass aerobic exercise to resistance training. Please outline the mode of exercise you considered for your inclusion criteria.
Results
Section 3.1
Line 3 –Is the mean and range of ages for females only, or for the total sample? It is not very clear in the current layout.
Section 3.7
Paragraph 2 Lines 6-9 – You state that PS plus exercise was not " for improving all mobility activities or physical function, but then go on to state this was not the case for walking capabilities or chair rise. Please restructures this sentence correctly to reflect this.
Table 1
In your selection criteria for your studies, you state that studies were only included if the adults in the study were aged over 60 years of age (ie criteria number 6 “participants were institutionalized residents or community-dwelled elders who aged more than 60 years and had high sarcopenia or frailty risks”). However, in table 1, reference [34] (Arciero et al., 2014) has been included, where the average age for adults is 47.0, 52.0 and 50.0. This then violates the inclusion criteria of your study design and ultimately invalidates your findings as all of your statistics would need to be reperformed and reanalysed with this study omitted.
Additionally, having read the full-text article of Arciero et al. (2014), there is no mention within their study of the participants having sarcopenia or at a high frailty risk, again, inclusion criteria for your study. In fact, Arciero et al. (2014) examines older adults classified as overweight or obese. In lieu of this, and without checking every other article you have presented, I have serious doubts about the accuracy of your findings.
Minor
The manuscript requires substantial editing to check for correct usage of the English language and formatting for consistency in terminology and writing style. The below are the formatting changes that I have identified, though this is by no means exhaustive.
Please be consistent with your abbreviations, either when introducing them or defining them. For example, you change between MSE and ET, and even use Ex, to describe exercise.
Additionally, the figure quality could be of a higher resolution. Please export your figures from into a vector-based format (such as PDF) to negate this problem.
Another issue I have highlighted is you state that there were no associations between certain variables against ALL factors, but then proceed to give exceptions to this (see paragraphs 2 and 3 of section 3.7 as an example of what I mean). As there are exceptions, such a statement can’t be made. I would report significant associations first, and then state that there were no other significant associations observed thereafter. Please restructure your statements where appropriate.
Introduction
Paragraph 1
Line 2 - I’m not sure I like the word “faired”. I think “impaired" would fit better.
Line 6 - “Elder population” should be “elder populations”.
Paragraph 2
Line 5-6 – Delete “that” and hyphenate “intervention induced”.
Line 8 – “…function recovery…” doesn’t make sense in the current context.
Line 11 – “elder population” should be “older populations”.
Methods
Section 2.1
Line 9-10 – “Reviewers” should be changed to “authors”.
Section 2.3
Line 13 – “in vivo and vivo” should be changed to “in vitro and in vivo” and italicised as this is Latin.
Section 2.6
Paragraph 3 Line 2 – Would long term follow-up not be ≥12 months, as medium term was ≥6 months but<12 months?
Paragraph 5 Line 5 – The manufacturing information for SPSS has already been stated in the previous paragraph so it is not required again. Please delete.
Results
Section 3.1
Line 2 – Change “those were published from 1992 until 2019” to “which were published between 1992 and 2019” as this reads better.
Line 3 – Change “Of all participant” to “From the total sample”.
Section 3.3
Line 2 – Add “the” between “protein, majority”.
Line 3 – Change “with” to “where”.
Section 3.4
Line 1 – “ET” has yet to be defined. Besides, should this not be muscle strengthening exercise (MSE)?
Section 3.5
Line 1- The sentence regarding methodological quality is fragmented. You could state “The methodological quality of the 33 RCTs were classified as follows; 20 as high…” and so forth.
Line 4 – “Interrater” should be hyphenated to “Inter-rater”.
Section 3.6
Line 2 – “ALM” has yet to be defined in the text.
Line 5 – “ET” should be “MSE” should it not? Also, delete “There was”.
Figure 2
Line 1 – Exercise training has now been defined as Ex! What happened to ET, and MSE for that matter, as your overall definition of exercise? Please be consistent.
What does 18.1.1, 18.1.2 and 18.1.3 refer to?
Section 3.7
Paragraph 1 Line 1 – Change “of the” to “in”.
Paragraph 2 Line 1-6 – You have changed from numerical to written reporting of the number of RCTs and comparisons. For example, it changes from “11 RCTs (17 comparisons)” to “five RCTs (eight comparisons)”. Please be consistent.
Paragraph 3 Line 1-4 – This sentence doesn’t make sense with the word “excepting” as a connector word. Moreover, you can’t say all factors generally had no significant effect, but then go on to violate this by presenting cases which showed significant associations. Please restructure.
Figure 3
Line 1 & 5 – “Metaregression” should be hyphenated to “meta-regression”. The same applies to figures 4 and 5 also.
“Side effects and compliance” – Should this not be section 3.9? Moreover, on line 2 after this, why are exercise training and protein supplementation now stated in full? Please be consistent.
Section 3.9
This should now be section 3.10.
Table 3
Intervention duration should be 12-23 weeks. Same applies to table 4.
References
Please check that all journal names in your references have been abbreviated to conform with the citation style of Nutrients.
Author Response
[Nutrients]
Manuscript ID: nutrients-538306
Author's Reply to the Review Report (Reviewer 2)
Comments and Suggestions for Authors
The authors present a carefully performed meta-analysis and systematic review of the data examining the efficacy of protein supplementation and exercise in the improvement of muscle mass, muscle function, and physical function in older, sarcopenic adults. Whilst the study has been carefully analysed, there is a significant flaw within your study which has invalidated your findings.
In table 1, and throughout the results, you reference and use Arciero et al. (2014) as one of your 33 RCTs you have analysed. However, this study invalidates your selection criteria as not only are the participants of this study below the age of 60, they are classified as overweight or obese in the paper, with no reference to these adults as having sarcopenia and at a high risk of frailty. This study violates your inclusion criteria, and as such should not have been included. Therefore, the analyses, subgroup analyses and conclusions of your study are not valid and would need to be reperformed and reanalysed. However, as such a study was included without the participants being classified as sarcopenic or frail, I can no longer trust whether the remaining 32 articles are in keeping with the inclusion criteria.
Response
Thank you for your comprehensive review and constructive comments. We have defined the study participants in Section 2.3 as follows:
“the study enrolled participants with mean age ≥ 60 years; the participants were hospitalized, institutionalized, or community-dwelling elderly individuals and with a high risk of sarcopenia or frailty and physical limitations.”
Accordingly, we removed the studies (including Arciero et al. 2014) that enrolled sedentary older adults and reperformed all analyses. The results are presented in the revised manuscript.
Additionally, the rationale for the study is weak. You state that previous meta-analyses have examined protein supplementation and exercise in relation to sarcopenia, but do not make explicitly clear how your study is novel or different. In my mind, the novelty is the combined meta-analyses/systematic review approach and that you are examining (or should be examining) sarcopenic and frail older adults who had been enrolled in randomised controlled trials.
Response
We have added statements to clarify the novelty of this study in the last paragraph of the Introduction section as follows:
“Previous systematic reviews and meta-analyses have investigated the effects of PS+ MSE on either sarcopenic or frail elderly populations; however, the combined meta-analysis approach in elderly adults with sarcopenia and frailty has not yet been confirmed. This study examined the combined effects of PS+ MSE in elderly adults who have high risks of sarcopenia and frailty.”
Finally, your text requires substantial editing for consistency of terminology (see below under minor), grammar, and usage of the English language.
Response
We have submitted our revised manuscript to avail of a professional editing service provided by Wallace Academic Editing.
Introduction
Paragraph 2
Line 2 – “Acute and chronic conditions” is vague. Do you mean with acute sarcopenia, chronic sarcopenia, or some other kind of condition?
Response
We have revised the statement as follows:
“Various nutrient interventions, exercise therapies or a combination of both are advised to prevent sarcopenia or frailty in elderly individuals [8-12],…”
Line 6 – Is “function” physical function or muscle function? I’d be clear here and throughout.
Response
We have revised the statement as follows:
“However, whether intervention-induced changes in muscle mass contribute to strength gain and physical mobility improvement after PS+MSE remains unclear.”
Line 5-8 – This section is unclear. You state that it is unclear whether a combination of PS and MSE has an effect on strength and function, yet in the following sentence, you list meta-analyses which have shown exactly that. Also, are you saying that these meta-analyses have examined muscle strength and physical functioning independently as per the “or” in line 8? If so, this needs to be made clearer.
Response
We have revised the statement as follows:
“However, whether intervention-induced changes in muscle mass contribute to strength gain and physical mobility improvement after PS+MSE remains unclear. An individual with lean mass (LM) gain exhibits improved physical performance, and several previous meta-analysis studies have reported that an increase in LM is accompanied by significant strength gain [14, 16-19] or improvements in physical functioning [14, 18] after PS+MSE; however, other authors have reported conflicting results of such synergetic improvements in LM and strength [13, 20] or physical function [17, 19].”
This leads to my next point of novelty. If there are meta-analyses existing to have shown improvements in muscular strength and physical function in sarcopenic adults following PS and MSE, then how is your study novel? Reference 15 in particular appears to be very similar. Make it explicitly clear in your introduction how your study is different. The fact you have used RCT’s in adults with high sarcopenia and frailty risk makes this novel, but has not been made clear in the introduction.
Response
We have added statements to clarify the novelty of this study in the last paragraph of the Introduction section as follows:
“Previous systematic reviews and meta-analyses have investigated the effects of PS+ MSE on either sarcopenic or frail elderly populations; however, the combined meta-analysis approach in elderly adults with sarcopenia and frailty has not yet been confirmed. This study examined the combined effects of PS+ MSE in elderly adults who have high risks of sarcopenia and frailty.”
General
As you are examining studies which have used older adults with a high sarcopenia or frailty risk, some indices of classifying older adults as clinically having sarcopenia or high frailty risk needs to be included in the introduction. The revised European Working Group on Sarcopenia consensus (EWGSOP2) would be a good place to start (Cruz-Jentoft et al., 2019; Age Ageing, 48[1], 16-31).
Response
We have added statements in the first paragraph of the Introduction section as follows:
“In addition, the indices for classifying older adults as clinically having sarcopenia [4] or high frailty risk [5] have been established among which low muscle strength and poor physical performance, such as slow walking speed, are common risk factors.”
Method
Section 2.3
As you are examining MSE, the type of muscle strengthening exercise should be considered in more detail in your inclusion criteria. In your results (section 3.4), you state exercise training, which is very broad and could encompass aerobic exercise to resistance training. Please outline the mode of exercise you considered for your inclusion criteria.
Response
We have added statements to outline the exercise type for inclusion criteria in Section 2.3 as follows:
“(4) exercise types included resistance training or a multicomponent exercise regime that consisted of MSE, aerobic exercise, balance training, and physical activity training;”
We have also added statements to outline the mode of exercise used by the included studies in Section 3.4 as follows:
“Regarding the mode of exercise, seven RCTs used resistance exercise training [37, 41, 43, 48, 51, 52, 54], 11 RCTs used multicomponent exercise regime, and one used aerobic training with weighted walking [53].”
Results
Section 3.1
Line 3 –Is the mean and range of ages for females only, or for the total sample? It is not very clear in the current layout.
Response
We have revised the statement as follows:
“A sample consisting of a total of 1888 participants with a mean (range) age of 78.7 (64.0-89.2) years was enrolled.”
Section 3.7
Paragraph 2 Lines 6-9 – You state that PS plus exercise was not " for improving all mobility activities or physical function, but then go on to state this was not the case for walking capabilities or chair rise. Please restructures this sentence correctly to reflect this.
Response
We have revised the statement as follows:
“Significant effects favoring PS+MSE were observed on walking capability (SMD = 0.33, 95% CI: 0.14-0.52; P = 0.0006; I2 = 39%; Figure 2 and Figures S5). No significant effects were identified in other mobility (Figure 2 and Figures S6-S10).”
Table 1
In your selection criteria for your studies, you state that studies were only included if the adults in the study were aged over 60 years of age (ie criteria number 6 “participants were institutionalized residents or community-dwelled elders who aged more than 60 years and had high sarcopenia or frailty risks”). However, in table 1, reference [34] (Arciero et al., 2014) has been included, where the average age for adults is 47.0, 52.0 and 50.0. This then violates the inclusion criteria of your study design and ultimately invalidates your findings as all of your statistics would need to be reperformed and reanalysed with this study omitted.
Additionally, having read the full-text article of Arciero et al. (2014), there is no mention within their study of the participants having sarcopenia or at a high frailty risk, again, inclusion criteria for your study. In fact, Arciero et al. (2014) examines older adults classified as overweight or obese. In lieu of this, and without checking every other article you have presented, I have serious doubts about the accuracy of your findings.
Response
Thank you for your comprehensive review and constructive comments. We have defined the participants in Section 2.3 as follows:
“(6) the study enrolled participants with mean age ≥ 60 years; the participants were hospitalized, institutionalized, or community-dwelling elderly individuals and with a high risk of sarcopenia or frailty and physical limitations.”
Accordingly, we removed the studies (including Arciero et al. 2014) that enrolled sedentary older adults and reperformed all analyses; the results are presented in the revised manuscript.
Minor
The manuscript requires substantial editing to check for correct usage of the English language and formatting for consistency in terminology and writing style. The below are the formatting changes that I have identified, though this is by no means exhaustive.
Response
We have submitted our revised manuscript to a professional editing service by Wallace Academic Editing.
Please be consistent with your abbreviations, either when introducing them or defining them. For example, you change between MSE and ET, and even use Ex, to describe exercise.
Response
We have used the term “MSE” to describe exercise throughout the revised manuscript.
Additionally, the figure quality could be of a higher resolution. Please export your figures from into a vector-based format (such as PDF) to negate this problem.
Response
We have modified the figure quality to obtain a higher resolution. We have also provided the PDF file of each figure on the online submission system.
Another issue I have highlighted is you state that there were no associations between certain variables against ALL factors, but then proceed to give exceptions to this (see paragraphs 2 and 3 of section 3.7 as an example of what I mean). As there are exceptions, such a statement can’t be made. I would report significant associations first, and then state that there were no other significant associations observed thereafter. Please restructure your statements where appropriate.
Response
We have rephrased paragraphs 2 and 3 in Section 3.7 as follows:
“Significant effects favoring PS+MSE were observed on walking capability (SMD = 0.33, 95% CI: 0.14-0.52; P = 0.0006; I2 = 39%; Figure 2 and Figures S5). No significant effects were identified in other mobility (Figure 2 and Figures S6-S10).
The results of subgroup analyses showed that the control group types exhibited effects on leg strength (I2 = 92.5%, P < 0.00001) and chair-rise scores (I2 = 70.1%, P = 0.04; Table 4). In addition, subgroup analyses for leg strength showed significant subgroup differences between participant types (I2 = 85.2%, P = 0.009) as well as among participant conditions (I2 = 88.7%, P = 0.0001) and intervention periods (I2 = 90.5%, P < 0.0001; Table 4). The institutionalized elderly participants with frailty exhibited a greater change in leg strength in response to PS+MSE with a greater SMD on leg strength by 1.02 (P < 0.00001) than the community-dwelled peers (SMD = 0.56, P = 0.0001). No other factor was found to affect subgroup heterogeneity for leg strength, walking capability, and chair-rise test (all P > 0.05) (Table 4).”
Introduction
Paragraph 1
Line 2 - I’m not sure I like the word “faired”. I think “impaired" would fit better.
Response
We have revised the statement as follows:
“Aging is associated with muscle attenuation, which may contribute to common characteristics of muscle weakness and impaired physical mobility observed in elderly individuals at high risks of sarcopenia and frailty [1-3].”
Line 6 - “Elder population” should be “elder populations”.
Response
We have revised the statement as follows:
“because low levels of lean mass or appendicular skeletal mass are closely associated with physical difficulty and poor health status among elderly patients [6, 7].”
Paragraph 2
Line 5-6 – Delete “that” and hyphenate “intervention induced”.
Response
We have revised the statement as follows:
“However, whether intervention-induced changes in muscle mass contribute to strength gain and physical mobility improvement after PS+MSE remains unclear.”
Line 8 – “…function recovery…” doesn’t make sense in the current context.
Response
We have revised the statement as follows:
“… several previous meta-analysis studies have reported that an increase in LM is accompanied by significant strength gain [14, 16-19] or improvements in physical functioning [14, 18] after PS+MSE;”
Line 11 – “elder population” should be “older populations”.
Response
We have revised the statement as follows:
“Given that low muscle mass is a well-established factor associated with strength loss and mobility limitations in elderly populations [7, 21]…”
Methods
Section 2.1
Line 9-10 – “Reviewers” should be changed to “authors”.
Response
We have revised the statement as follows:
“Two authors (CDL and HCC) independently searched for relevant articles, screened them, and extracted data. Any disagreement between the authors were resolved through a consensus in which the other team members (THL and SWH) acted as arbitrators.”
Section 2.3
Line 13 – “in vivo and vivo” should be changed to “in vitro and in vivo” and italicised as this is Latin.
Response
We have revised the statement as follows:
“(1) the trial was conducted in vitro or in vivo in an animal model;…”
Section 2.6
Paragraph 3 Line 2 – Would long term follow-up not be ≥12 months, as medium term was ≥6 months but<12 months?
Response
Thank you. We revised the statement as follows:
“The duration of follow up (FU) was assessed and defined as immediate (<3 months), short term (≥3 months, <6 months), medium term (≥6 months, <12 months), and long term (≥12 months).”
Paragraph 5 Line 5 – The manufacturing information for SPSS has already been stated in the previous paragraph so it is not required again. Please delete.
Response
We have revised the statement as follows:
“…; the analysis was controlled for age, methodological design, and follow-up duration.”
Results
Section 3.1
Line 2 – Change “those were published from 1992 until 2019” to “which were published between 1992 and 2019” as this reads better.
Response
We have revised the statement as follows:
“The final sample consisted of 19 RCTs [31, 37-54], which were published between 1994 and 2019.”
Line 3 – Change “Of all participant” to “From the total sample”.
Response
We have revised the statement as follows:
“From the total sample, 738 patients received a protein-type supplement in combination with MSE,…”
Section 3.3
Line 2 – Add “the” between “protein, majority”.
Line 3 – Change “with” to “where”.
Response
We have revised the statement as follows:
“Regarding the amount of protein, the majority of the included RCTs employed daily PS with amounts of extra protein ranging from 3.0 to 40.8 g/d. Three RCTs used PS of<10.0 g/d [31, 46, 47] …”
Section 3.4
Line 1 – “ET” has yet to be defined. Besides, should this not be muscle strengthening exercise (MSE)?
Response
We have revised the statement as follows:
“A summary of protocols for MSE is presented in Table 1. Regarding the mode of exercise, seven RCTs used resistance exercise training [37, 41, 43, 48, 51, 52, 54], 11 RCTs used multicomponent exercise regime, and one used aerobic training with weighted walking [53] …”
Section 3.5
Line 1- The sentence regarding methodological quality is fragmented. You could state “The methodological quality of the 33 RCTs were classified as follows; 20 as high…” and so forth.
Response
We have revised the statement as follows:
“Among the 19 RCTs, the methodological quality of 12 was high [37-41, 43, 45, 47, 48, 50, 51, 54] and that of the other seven was medium [31, 42, 44, 46, 49, 52, 53], with a median PEDro score of 7/10 (range 5/10 to 9/10).”
Line 4 – “Interrater” should be hyphenated to “Inter-rater”.
Response
We have revised the statement as follows:
“The interrater reliability associated with the cumulative PEDro score was acceptable with an intraclass correlation coefficient of 0.93 (95% CI: 0.82–0.97).”
Section 3.6
Line 2 – “ALM” has yet to be defined in the text.
Response
We have defined “ALM” in Section 2.3 as follows:
“(7) the study reported the primary outcome measures of muscle mass or sarcopenia indices, including lean body mass (LBM), fat-free mass, appendicular lean mass (ALM),”
Line 5 – “ET” should be “MSE” should it not? Also, delete “There was”.
Response
We have revised the statement as follows:
“Results of meta-analyses showed significant short-term (SMD = 0.71, P < 0.00001) and medium-term (SMD = 0.56, P = 0.02) effects on LBM as well as on ALM in favor of PS+MSE (Figure 2 and Figures S1-S2). The evidence showed an overall effect on LBM…”
Figure 2
Line 1 – Exercise training has now been defined as Ex! What happened to ET, and MSE for that matter, as your overall definition of exercise? Please be consistent.
Response
We have used the term “MSE” to describe exercise throughout the revised manuscript.
What does 18.1.1, 18.1.2 and 18.1.3 refer to?
Response
Thank you. We removed 18.1.1, 18.1.2 and 18.1.3 from Figure 2.
Section 3.7
Paragraph 1 Line 1 – Change “of the” to “in”.
Response
Thank you. We revised the statement as follows:
“Changes in the handgrip and leg strength were reported by 6 RCTs (13 comparisons) ……”
Paragraph 2 Line 1-6 – You have changed from numerical to written reporting of the number of RCTs and comparisons. For example, it changes from “11 RCTs (17 comparisons)” to “five RCTs (eight comparisons)”. Please be consistent.
Response
We have revised the statement as follows:
“The treatment effect of PS+MSE on physical function was assessed using several mobility tests, including walking capability by 18 RCTs (23 comparisons) [31, 37-39, 41, 43, 44, 47, 48, 51, 52], chair-rise test by seven RCTs (13 comparisons) [37-39, 41, 48, 51, 52], timed up-and-go by two RCTs (three comparisons) [38, 48], stair-climb test by three RCTs (five comparisons) [38, 39, 43], short physical performance battery by three RCTs (three comparisons) [41, 44, 51], and single leg stance by one RCT (six comparisons) [52].”
Paragraph 3 Line 1-4 – This sentence doesn’t make sense with the word “excepting” as a connector word. Moreover, you can’t say all factors generally had no significant effect, but then go on to violate this by presenting cases which showed significant associations. Please restructure.
Response
We have rephrased the Paragraph 3 in Section 3.7 as follows:
“The results of subgroup analyses showed that the control group types exhibited effects on leg strength (I2 = 92.5%, P < 0.00001) and chair-rise scores (I2 = 70.1%, P = 0.04; Table 4). In addition, subgroup analyses for leg strength showed significant subgroup differences between participant types (I2 = 85.2%, P = 0.009) as well as among participant conditions (I2 = 88.7%, P = 0.0001) and intervention periods (I2 = 90.5%, P < 0.0001; Table 4). The institutionalized elderly participants exhibited a greater change in leg strength in response to PS+MSE with a greater SMD on leg strength by 1.02 (P < 0.00001) than the community-dwelled peers (SMD = 0.56, P = 0.0001). No other factor was found to affect subgroup heterogeneity for leg strength, walking capability, and chair-rise test (all P > 0.05) (Table 4).”
Figure 3
Line 1 & 5 – “Metaregression” should be hyphenated to “meta-regression”. The same applies to figures 4 and 5 also.
Response
“Metaregression” has been hyphenated to “meta-regression” throughout the revised manuscript.
“Side effects and compliance” – Should this not be section 3.9? Moreover, on line 2 after this, why are exercise training and protein supplementation now stated in full? Please be consistent.
Response
We have revised the statement as follows:
“The compliance of resistance-based and multicomponent-based MSE was reported as 84%–100% by six RCTs [41, 43, 48, 51, 52, 54] and 44%–81% by six RCTs [38-40, 42, 44, 47], respectively (Table 1). The compliance of PS was reported as 44%–100% by 13 RCTs [37-40, 42-45, 48, 50-53] (Table 1).”
Section 3.9
This should now be section 3.10.
Table 3
Intervention duration should be 12-23 weeks. Same applies to table 4.
Response
We have revised the Intervention duration to be 12-23 weeks in Tables 3 and 4.
References
Please check that all journal names in your references have been abbreviated to conform with the citation style of Nutrients.
Response
We have confirmed all journal names in the cited references to conform to the citation style of Nutrients.

Round 2
Reviewer 2 Report
To Liao and colleagues,
Thank you for taking the time to systematically address each of my concerns regarding your original submission of your manuscript. I appreciate that I highlighted man y issues with the first submission, but your responses are commendable. The manuscript has been improved as a result, and with the omission of Arciero et al. (2014) from the analyses, the study can now be considered a valid contribution. In particular, having the manuscript proof read has been beneficial.
I still have a minor issue with your manuscript:
Figure 2 – The numbers 18.1.1, 18.1.2 and 18.1.3 are still in the figure despite stating these have been removed.
Author Response
To Liao and colleagues,
Thank you for taking the time to systematically address each of my concerns regarding your original submission of your manuscript. I appreciate that I highlighted man y issues with the first submission, but your responses are commendable. The manuscript has been improved as a result, and with the omission of Arciero et al. (2014) from the analyses, the study can now be considered a valid contribution. In particular, having the manuscript proof read has been beneficial.
I still have a minor issue with your manuscript:
Figure 2 – The numbers 18.1.1, 18.1.2 and 18.1.3 are still in the figure despite stating these have been removed.
Response
Thank you for your comprehensive review and constructive comments. We removed the numbers 18.1.1, 18.1.2 and 18.1.3 from Figure 2 in the first-round revised manuscript. We now confirm it and resubmitted a second-round revised manuscript here for review.
